# Measuring the Benefits of Mass Vaccination Programs in the United States

**DOI:** 10.3390/vaccines8040561

**Published:** 2020-09-29

**Authors:** Hector Magno, Beatrice Golomb

**Affiliations:** 1Independent Computer Scientist, Orange County, CA 92677, USA; hmagno@protonmail.com; 2Department of Medicine, UC San Diego School of Medicine, La Jolla, CA 92093, USA

**Keywords:** vaccination, disease, mortality, disability, risk

## Abstract

Since the late 1940s, mass vaccination programs in the USA have contributed to the significantly reduced morbidity and mortality of infectious diseases. To assist the evaluation of the benefits of mass vaccination programs, the number of individuals who would have suffered death or permanent disability in the USA in 2014, had mass vaccination never been implemented, was estimated for measles, mumps, rubella, tetanus, diphtheria, pertussis, polio, *Haemophilus influenzae* type b (Hib), hepatitis B, varicella, and human papillomavirus (HPV). The estimates accounted for mortality and morbidity trends observed for these infections prior to mass vaccination and the impact of advances in standard of living and health care. The estimates also considered populations with and without known factors leading to an elevated risk of permanent injury from infection. Mass vaccination prevented an estimated 20 million infections and 12,000 deaths and permanent disabilities in 2014, including 10,800 deaths and permanent disabilities in persons at elevated risk. Though 9000 of the estimated prevented deaths were from liver cirrhosis and cancer, mass vaccination programs have not, at this point, shown empirical impacts on the prevalence of those conditions. Future studies can refine these estimates, assess the impact of adjusting estimation assumptions, and consider additional risk factors that lead to heightened risk of permanent harm from infection.

## 1. Introduction

To measure the benefit of a mass vaccination program targeting an infectious disease, it is useful to assess what the risk of death or permanent injury would be from the disease in the absence of the mass vaccination program. There is an abundance of medical literature detailing the risks associated with infectious diseases; however, the information is scattered through dozens of sources that are often lengthy and consider only a narrow scope of the risks involved. For example, some sources describe the symptoms of a disease without specifying how many patients fully recover [1]; other sources describe the number of deaths from an infection without addressing permanent disability in survivors [2,3]. Moreover, some sources do not account for the pre-vaccine rates of decline in mortality for some infectious diseases [3,4,5]. We tried to address these challenges in our estimates.

## 2. Materials and Methods

We calculated rates of deaths and disabilities from infectious diseases, had mass vaccination never been implemented, using data principally from reports of the Centers for Disease Control and Prevention (CDC), complemented by reports from other federal entities such as the US Bureau of the Census and the US Public Health Service. We relied on data recorded in scientific journals (e.g., The Journal of the American Medical Association, Pediatrics, The Journal of Infectious Diseases, The New England Journal of Medicine, and The Journal of Clinical Oncology) in cases when data from government sources were unavailable or incomplete. For example, although the impact of risk factors for many diseases is considered in CDC data, there are instances when measurements of risk factor relationships to outcomes are not provided. Other examples of information that is not always available in government records include estimates of the number of unreported cases and of permanent disability from certain diseases. In addition, when US data for measurements were unavailable, we relied on data from other developed countries.

Because we researched and gathered data for our estimates in 2016, when the latest available CDC mortality data were from 2014, we projected our estimates to 2014. When calculating the rates of deaths and disabilities corresponding to an infectious disease, we considered the trend in deaths and disabilities during a range of years just prior to the licensing of a vaccine or at the start of a nationwide mass vaccination program targeting the disease. This range of years is referred to as the “reference years.” The duration of each range was chosen by using the number of years between relative peaks of incidence of each disease.

Though <100% of the population is vaccinated and vaccines are <100% effective, mass vaccination programs have contributed to the significantly reduced morbidity and mortality of infectious diseases. There were fewer than two dozen deaths from diphtheria, tetanus, pertussis, polio, measles, mumps, rubella, *Haemophilus influenzae* type b (Hib), or varicella in 2014. Therefore, we counted all estimated cases projected to 2014 in the absence of mass vaccination as preventable.

We employed certain assumptions in our estimates. For example, we presumed that each vaccine neither reduced nor enhanced vulnerability to the incidence or outcomes of diseases that were not targeted by the vaccine. Throughout our text, we strive to be explicit whenever an assumption was made.

The rates of deaths or disabilities corresponding to various infectious diseases were computed using a denominator of 307 million individuals <80 years of age in 2014. We chose that age group because the life expectancy in 2014 was 79 years. In instances where age-specific counts of cases of infection were not available, case counts from the entire US population (319 million) were used. Moreover, we considered the broader population <80 years of age rather than the population of children, both because mass vaccination programs are intended to provide lifetime immunity to infection and because protecting against infection averts late complications of infection. For example, the polio mass vaccination program affected not only permanent injury from polio in children but also permanent injury from post-polio syndrome in adults.

To simplify and reduce the length of the report, we omitted analyses of certain mass vaccination programs that had marginal or unclear impacts on mortality. We excluded rotavirus and hepatitis A because each of those infections caused fewer than 100 annual pre-vaccine deaths [2,6]. We excluded influenza because its mass vaccination program has not made a clear impact on the trend of its pre-vaccine mortality rate [7,8,9]. Though there is an effective vaccine targeting meningococcal disease, the trend in the mortality rate for that infection after the introduction of its mass vaccination program has resembled the pre-vaccine trend [10]. Consequently, we excluded meningococcal disease because the matching mortality rate trends suggested that fewer than 100 annual deaths were prevented. We excluded pneumococcal disease for a similar reason. The CDC’s estimated decline in pneumococcal disease mortality after the introduction of the mass vaccination program matched the rate of decline in mortality from all pneumonia, including pneumonia caused by pathogens that are not targeted by the vaccine. From 2000 to 2009, the CDC estimated that pneumococcal disease mortality declined from 2.3 per 100,000 population to 1.6, a 30% decline [11,12]. During the same time period, the mortality rate of pneumonia from all causes declined from 22.6 to 16.6, a 27% decline [13]. The similar rates of decline suggested that the mass vaccination program prevented fewer than 100 deaths among individuals <80 years of age. 

When accounting for risk factors leading to an elevated risk of permanent injury from an infection, we included only those factors that were observed in a high fraction of cases of permanent injury from the infection. To simplify the report, risk factors that were present in a small fraction of such cases were excluded if accounting for those factors resulted in a rate of permanent injury that lay within the 95% confidence interval of the rate computed without those factors (i.e., did not make a statistically significant impact on the rate).

## 3. Results

It was estimated that 20 million infections and 12,000 deaths and permanent disabilities may have occurred in 2014 in the absence of mass vaccination, with 10,800 deaths and disabilities among individuals who have conditions or behaviors that would put them at higher risk of such outcomes and 1200 deaths and disabilities among persons without those conditions or behaviors. Table 1 and Table 2 show the aggregated results for the infectious diseases examined in this report.

The following is a discussion of each disease. Recall that the “reference years” refer to the time period before the introduction of the corresponding mass vaccination program. Using data recorded during these years, we derived estimates of the expected number of deaths and permanent disabilities from each disease had mass vaccination not been introduced.

### 3.1. Measles

During the reference years of 1959–1962, before the introduction of mass vaccination, there were four million annual measles cases (equal in size to the birth cohort; Appendix A) that resulted in 402 deaths [2] mostly among the population <10 years of age [14]. Because the birth cohorts in the 1960s and in 2014 were the same size and the number of susceptible children was also the same [15,16], we estimated that these values would remain unchanged in the absence of mass vaccination. We also estimated 106 additional cases of measles that resulted in residual neurologic damage from complications of measles including measles encephalitis and subacute sclerosing panencephalitis (Appendix A).

Individuals with low levels of vitamin A are significantly more likely to suffer death or permanent disability from measles [17,18]; 92% of the most severe measles cases have had low levels of vitamin A (Appendix A) [19]. Therefore, we calculated 467 (=92% of 508) measles deaths and permanent disabilities at elevated risk.

Though the pre-vaccine measles mortality rate declined from 14.1 to 0.2 per 100,000 people (Appendix A), the measles fatalities recorded in the 1980s and 1990s suggested that the pre-vaccine decline may not have continued as rapidly in the absence of mass vaccination [17]. Consequently, we assumed that the measles mortality rate would have remained unchanged from the reference years.

### 3.2. Mumps

During the reference years of 1963–1966, before the introduction of mass vaccination, there were four million annual mumps cases (equal in size to the birth cohort; Appendix A) that resulted in 43 deaths [2], mostly among the population <30 years of age [20]. Because the birth cohorts in the 1960s and in 2014 were the same size and the number of susceptible children was also the same [15,16], we estimated that these values would remain unchanged in the absence of mass vaccination. We also estimated 11 additional cases of mumps resulting in permanent impaired hearing and 7 additional cases of mumps resulting in permanent impaired fertility (Appendix A).

### 3.3. Rubella (German Measles)

During the reference years of 1960–1968, before the introduction of mass vaccination, there were four million annual rubella cases (equal in size to the birth cohort; Appendix A) that resulted in 19 deaths [2]. CDC analyses of rubella have shown that permanent disability in rubella survivors was very rare [21]. However, mass vaccination was adopted because of congenital rubella syndrome (CRS) [21], which posed a threat to those infants whose mothers were infected by rubella during the first two trimesters of pregnancy. Therefore, we additionally considered the number of babies that contracted rubella in utero.

Using the government tracking of cases of rubella and CRS, we estimated that during the nine reference years, there were 1484 cases of CRS or 165 cases annually (Appendix A). Of those cases, we calculated 140 (=85% of 165) that resulted in death or permanent disability [21].

Because the birth cohorts in the 1960s and in 2014 were the same size and the number of susceptible children was also the same [15,16], we estimated that these values would remain unchanged in the absence of mass vaccination. Furthermore, most women in the 1960s contracted rubella before childbearing age [22], and the typical childbearing age of women in 2014 was greater than it was in the 1960s. Therefore, the estimated 165 cases of CRS for 2014 may be an overestimation.

### 3.4. Tetanus

Using the government tracking of cases of tetanus, we calculated that, during the reference years of 1943–1945, before the introduction of mass vaccination, one case of tetanus occurred in every 180,000 people (Appendix A). To estimate the number of tetanus cases for 2014 in the absence of mass vaccination, we multiplied the pre-vaccine incidence ratio of 1 in 180,000 by 2014′s population (319 million) to obtain 1800 cases of tetanus.

CDC analyses of tetanus have shown that permanent disability in tetanus survivors was very rare [23]. As for deaths from tetanus, we used the most recent case fatality rates recorded among unvaccinated populations to account for significant improvements in health care and other factors influencing disease outcomes since the 1940s. From 2001 to 2008, we calculated a case fatality rate of 6.3% among unvaccinated individuals <80 years of age (Appendix A). We multiplied the estimated 1800 cases of tetanus for 2014 by the case fatality rate of 6.3% to obtain 113 fatal cases.

The pre-vaccine decline in the tetanus mortality rate from 2.4 to 0.5 tetanus deaths per 100,000 people (Appendix A) provided additional support for the projected decline in annual tetanus deaths from 626 in the 1940s (Appendix A) to 113 in 2014.

### 3.5. Diphtheria

During the reference years of 1879–1945, before the introduction of the national mass vaccination program, there was an exponential decline in diphtheria morbidity and mortality (Appendix A) [24,25,26]. This decline predated the introduction of antitoxin in the late 1890s, the introduction of toxin–antitoxin in the 1920s, and the gradual introduction of toxoid in the 1930s [27,28]. Furthermore, none of those events significantly altered the decline, suggesting that non-vaccine factors played important roles [27,28]. Laboratory testing revealed the protective effects of vitamin C [29], iron [30], and vitamin B3 [31] against diphtheria toxin. Studies also revealed that crowding and low levels of hygiene were associated with high incidences of diphtheria [32]. 

Because significant improvements in nutrition, sanitation, living conditions, and access to health care continued after the reference years, we estimated that the 65-year decline would have continued in the absence of the national mass vaccination program of the late 1940s, and we calculated 28 diphtheria deaths for 2014 (Appendix A). Because the prevalence and severity of diphtheria risk factors required to put an individual at elevated risk have not been measured, we did not attempt to segregate cases at high risk from our results.

CDC analyses of diphtheria have shown that permanent disability in diphtheria survivors was very rare [28]. Additionally, given a case fatality rate of 5% for diphtheria [28], we estimated 560 (=28/5%) diphtheria cases for 2014.

### 3.6. Pertussis (Whooping Cough)

Using the government tracking of cases of pertussis during the reference years of 1943–1945, before the introduction of mass vaccination, we calculated 235,000 reported pertussis cases out of a total 1.3 million cases for 2014 in the absence of mass vaccination (Appendix A).

CDC analyses of pertussis have shown that permanent disability in pertussis survivors was very rare [33]. As for deaths from pertussis, we used the most recent case fatality rates recorded among unvaccinated populations to account for significant improvements in health care and other factors influencing disease outcomes since the 1940s. From 2012 to 2014, we calculated a reported case fatality rate of 0.7% among unvaccinated infants <3 months of age (Appendix A). We estimated 4500 reported pertussis cases for 2014 in that age group (Appendix A) and multiplied by the reported case fatality rate of 0.7% to obtain 32 fatal cases. Because infants <3 months of age comprised 26% of all pertussis deaths during the reference years (Appendix A), we calculated 123 (=32/26%) pertussis deaths among individuals of all ages.

The pre-vaccine decline in the pertussis mortality rate from 16.1 to 1.3 pertussis deaths per 100,000 people (Appendix A) and the mortality rate of one in eight million recorded in Sweden in the absence of mass vaccination in the 1980s [34] provided additional support for the projected decline in annual pertussis deaths from 2300 in the 1940s (Appendix A) to 123 in 2014.

### 3.7. Polio

During the reference years of 1935–1954, before the introduction of mass vaccination, there were an estimated 7260 annual cases of paralytic poliomyelitis (Appendix A), of which 1136 resulted in death or permanent disability (Appendix A). Since 95% of all polio infections were unnoticed or asymptomatic, and less than 1% of cases were paralytic [35], we estimated at total of 36,000 (≈5% of (7260/1%)) noticeable annual cases of polio. To estimate the number of noticeable cases of polio for 2014 in the absence of mass vaccination, we multiplied the pre-vaccine incidence ratio of 1 in 4400 (≈36,000/160 million) by the 2014 population (319 million) to obtain 72,500 cases.

Individuals without tonsils or who do not rest after the onset of significant symptoms are more likely to suffer permanent disability or death from paralytic poliomyelitis [36,37]. We calculated 1149 permanent disabilities and deaths among individuals at elevated risk and 353 among individuals at normal risk for 2014 in the absence of mass vaccination (Appendix A).

### 3.8. Hib (Haemophilus Influenzae Type b)

During the reference years of 1980–1984, before the introduction of mass vaccination, most children acquired immunity to Hib by five years of age through asymptomatic infection. In this report, we only considered identifiable cases of Hib-invasive Hib cases [38].

Using invasive *H. influenzae* data from the reference years and the government tracking of cases of invasive *H. influenzae*, we estimated an annual total of 3400 cases of invasive Hib for 1994–2000 in the absence of mass vaccination (Appendix A). Of those cases, we calculated 330 resulting in death or permanent disability from meningitis, bacteremia, or epiglottitis (Appendix A).

Children who were breastfed exclusively for ≥13 weeks were 2.8 times less likely to contract invasive Hib (Appendix A). On this basis, we used our estimates of invasive Hib incidence and permanent injury for 1994–2000 to calculate estimates for 2014 in the absence of mass vaccination: 66 cases of permanent injury among children breastfed exclusively for ≥13 weeks and 208 cases of permanent injury among children breasted for <13 weeks (Appendix A).

To estimate the number of all cases of invasive Hib for 2014, we divided the 274 cases of death and permanent disability by the percentage of Hib cases that resulted in such outcomes—9.8% (=[60% × (11% + 5%)] + [15% × 1%]; Appendix A)—to obtain 2800 cases.

### 3.9. Hepatitis B

Using the government tracking of cases of hepatitis B during the reference years of 1988–1990, before the introduction of mass vaccination, we calculated 190,000 cases for 2014 in the absence of mass vaccination (Appendix A), including 300 cases of fatal fulminant hepatitis—nearly all of which occurred in adults and adolescents (Appendix A).

CDC analyses of hepatitis B have shown that permanent disability in hepatitis B survivors is very rare [39]. However, a portion of hepatitis B survivors can develop a chronic infection that can lead to fatal cirrhosis or liver cancer later in life, and 85% of those deaths occur in individuals <80 years of age [40]. Using government chronic hepatitis B data, we estimated 1100 infections in adults and adolescents and 1740 infections in children resulting in chronic infection that led to death before age 80 for 2014 (Appendix A).

Individuals at high risk of exposure are more likely to contract hepatitis B (Table 3). Of the estimated 1400 deaths among adults and adolescents for 2014 in the absence of mass vaccination (300 from fulminant hepatitis + 1100 from chronic infection), we calculated 1300 deaths among individuals at elevated risk (Appendix A). Of the estimated 1740 deaths from infections in childhood, we calculated 1734 among children at elevated risk (Appendix A). Combining these totals resulted in 3034 (=1300 + 1734) hepatitis B-related deaths among individuals at elevated risk and 106 (=100 + 6) deaths among individuals at normal risk.

### 3.10. Varicella (Chicken Pox)

During the reference years of 1991–1994, before the introduction of mass vaccination, there were four million annual varicella cases (equal in size to the birth cohort; Appendix A) that resulted in 101 deaths [2], mostly among the population ≥20 years of age [41]. Because the birth cohorts in the 1990s and in 2014 were the same size and the number of susceptible adults was also the same [16,42], we estimated that this value would remain unchanged in the absence of mass vaccination.

CDC analyses of varicella have shown that permanent disability in varicella survivors is very rare [43]. Though zoster (shingles) can occur later in life in individuals infected with varicella, death or permanent disability from zoster is also very rare [44]. Thus, we estimated no cases of varicella-related permanent disability for 2014.

### 3.11. HPV (Human Papillomavirus)

The CDC has estimated that there are 14 million annual cases of HPV of all types [45]. Of those 14 million cases, 20% (2.8 million) are targeted by vaccines [46]. Most HPV infections are unnoticed or asymptomatic. When there are symptoms from an HPV infection, such as genital warts, they very rarely cause death or permanent disability [45]. However, a small proportion of individuals infected with HPV can become persistently infected, and this condition can lead to various kinds of cancers later in life [45,47]. The first HPV vaccine was licensed in 2006, and the vaccination program targeted teenagers. Since HPV-attributable cancers rarely affect individuals <30 years of age, it will take at least another decade before it is possible for mass vaccination to have a measurable effect on the incidence of those cancers. Here, we consider HPV-attributable cancer statistics during the reference years of 2011–2014.

Table 4 contains a list of HPV-attributable cancers and factors that lead to an elevated risk of dying from them. Among women <80 years of age, we estimated 132 fatal HPV-attributable cancers occurring in women at normal risk and 5340 fatal cancers occurring in women at elevated risk (Appendix A). Among men <80 years of age, we estimated 66 fatal HPV-attributable cancers occurring in men at normal risk and 371 fatal cancers occurring in men at elevated risk (Appendix A). Combining these totals resulted in 198 (=132 + 66) HPV-related deaths among individuals at normal risk and 5711 (=5340 + 371) deaths among individuals at elevated risk.

## 4. Discussion

Based on population data for 2014, it was estimated that mass vaccination programs against measles, mumps, rubella, tetanus, diphtheria, pertussis, polio, Hib, hepatitis B, varicella, and HPV could prevent 20 million infections and 12,000 deaths and permanent disabilities annually.

Individuals who have conditions or behaviors that would put them at higher risk of permanent injury from infectious diseases (e.g., insufficient vitamin A, absence of tonsils, breastfed <13 weeks, injection-drug use, and smoking) were found to comprise 90% (≈10,800/12,000) of all the estimated cases of prevented death and permanent disability, with the remaining 1200 cases in persons at normal risk (or with risk factors excluded from this report). It is possible that the high risk conditions described in this report might expose individuals to permanent harm from other causes. More research in this arena would be useful.

Pre-vaccine declines in mortality rates recorded for measles, tetanus, diphtheria, and pertussis were not unique to those infections. In the early 20th century, significant declines in mortality rates were recorded for numerous infectious diseases that were not targeted by mass vaccination programs, such as those for tuberculosis, syphilis, typhoid fever, and dysentery [48]. The human immune system is evidently remarkably efficient when coupled with treatments for severe cases of diseases, such as antibiotics and when not hampered by factors like poor nutrition, poor sanitation, or limited access to health care. 

Mass vaccination programs are best known for preventing deaths and permanent disabilities that occur a relatively short time after infection. However, 75% (≈9000/12,000) of the estimated cases of death and permanent disability prevented in this report would be from conditions occurring much later in life—liver cirrhosis and cancer. Though hepatitis B and HPV are the causes of these conditions, the hepatitis B and HPV mass vaccination programs have not, at this point, shown empirical impacts on the prevalence of liver cirrhosis and cancer. In spite of the significant reduction in acute cases of hepatitis B, the prevalence of chronic hepatitis B has remained practically unchanged since 1976 [49]. As for the HPV vaccine, although the prevention of HPV infections that are necessary for the potential development of cancer has been observed [45], cancer protection has not yet been empirically documented and uncertainties remain. Among these, a minimum protective antibody titer has not been determined [45], and the duration of antibody response has only been measured for eight-to-nine years [50]. Since most HPV-attributable cancers occur in the population >50 years of age, it may be that to most successfully prevent HPV-attributable cancer, either an HPV vaccine needs to provide lifetime immunity or booster doses need to be introduced into the mass vaccination program.

This report had other limitations. The accuracy of our estimates depended on the quality of the available evidence concerning the risks and effects of the diseases, which could have been imperfect. In many instances, we projected to 2014 from statistics recorded decades earlier. The accuracy of such projections could have been affected by changes in the organisms targeted by vaccines, changes in host resilience, or changes in health care practices deviating from pre-vaccine trends, among other factors. In addition, inaccuracies can propagate from one estimate to another when an estimate is used to derive the other. For example, the case fatality rate of a disease is sometimes used to estimate the total number of cases of that disease based on its estimated number of fatalities. Another limitation was related to the potential aggregate impact of risk factors that were not considered in generating our estimates, either because they are (individually) less commonly implicated or are undiscovered or inadequately studied. The comprehensive consideration of such factors might shift more of the vaccine-averted deaths and disability to the high-risk category. The same limitations might apply to the long term effects of some of the diseases. It is also possible that relevant information was missed in our literature review. We sought to convey the challenges surrounding some of the available data in the discussion of each disease and have tried to be explicit about assumptions made. We explained our choices in relation to the application of pre-vaccine trends and in gauging expected disease outcomes as a function of whether an individual is at higher risk. Furthermore, this study only estimated the number of deaths and permanent disabilities prevented by mass vaccination programs. It did not consider similar outcomes that may be caused by these programs.

Despite these limitations, we believe this report employed the best processes among the available data and studies for estimating the numbers of deaths and permanent disabilities that would have occurred (here estimated for 2014) in the absence of mass vaccination programs. Though other studies have presented estimates of the benefits of mass vaccination programs, they have not accounted for disease risk factors, cases of nonfatal permanent disability, pre-vaccine trends in mortality, post-vaccine improvements in factors tied to disease outcomes resulting in improved case fatality rates in unvaccinated populations (such as improved nutrition, sanitation, hygiene, indoor temperature control, health care, and the treatment of disease), and adjustments of pre-vaccine estimates using data recorded after vaccine licensure [3,4,5]. Furthermore, some studies have not provided an explanation for the data used as the basis of their estimates [4,5]. We have tried to rectify these omissions in the present report.

## 5. Conclusions

Despite the decline in mortality rates of infectious diseases recorded since the late 19th century, the data in this report indicate that mass vaccination programs may still have prevented 20 million infections and 12,000 deaths and permanent disabilities in 2014. In addition, mass vaccination programs have reduced the burden on health services, hospitals, intensive care, and the economy caused by the diseases they target. Put another way, measuring the number of deaths and cases of permanent disability prevented by mass vaccination programs is not the only way to measure the benefit of those programs, as those benefits can also be measured by other outcomes such as hospitalizations or the economic burden associated with a disease. However, those outcomes may be more greatly influenced by a range of factors beyond the impact of the vaccine and the disease, such as shifts in approaches to and costs of hospitalization over time. Nonetheless, because such outcomes are generally a function of the morbidity and mortality of the disease, the data in this report might also be useful in generating estimates for those outcomes.

We believe this report provides a useful reference for the effect of mass vaccination programs on the most serious complications of the diseases they target. Future studies can seek to further refine these estimates, use these estimates in risk-benefit analyses, and assess how adjusting assumptions influences effect estimates.

## Figures and Tables

**Table 1 vaccines-08-00561-t001:** Estimated rates of death and permanent disability from various infectious diseases in the USA in the absence of mass vaccination among normal and high risk individuals <80 years of age, 2014.

Infection	Range of Reference Years Used for Estimates	Number of Cases (Morbidity) (A)	Population in 100,000 s (B)		Estimated Number of Cases of Death and Permanent Disability (C)		Rate: Deaths and Disabilities Per Number of Cases [C ÷ A]	Rate: Deaths and Disabilities Per 100,000 Population [C ÷ B] (95% C.I.)	
			Normal Risk ^a^	High Risk ^a^	Normal Risk ^a^	High Risk ^a^	All Risk	Normal Risk ^a^	High Risk ^a^
**Measles**	1959–1962	4,000,000	2920	150	41	467	0.01%	0.014 (0.010–0.018)	3.11 (2.83 to 3.40)
**Mumps**	1963–1966	4,000,000	3070	0	61	0	0.002%	0.020 (0.015–0.025)	0
**Rubella**	1960–1968	4,000,000	3000	70	19	140	0.004%	0.006 (0.003–0.009)	2.00 (1.67–2.33)
**Tetanus**	1943–1945	1800	3070	0	113	0	6.3%	0.037 (0.029–0.043)	0
**Diphtheria**	1879–1945	560	3070	0	28	0	5%	0.009 (0.006–0.012)	0
**Pertussis**	1943–1945	1,300,000	3070	0	123	0	0.009%	0.040 (0.033–0.047)	0
**Polio**	1935–1954	72,500	2480	590	353	1149	2.1%	0.142 (0.127–0.157)	1.95 (1.83–2.06)
**Hib**	1980–1984	2800	1440	1630	66	208	9.8%	0.046 (0.035–0.057)	0.13 (0.11–0.14)
**Hepatitis B**	1988–1990	190,000	2610	460	106	3034	1.7%	0.041 (0.033–0.048)	6.60 (6.36–6.83)
**Varicella**	1991–1994	4,000,000	3070	0	101	0	0.003%	0.033 (0.026–0.039)	0
**HPV**	2011–2014	2,800,000	1750	1320	198	5711	0.2%	0.113 (0.097–0.129)	4.33 (4.22–4.44)

^a^ “High risk” refers to individuals with specified factors linked to an elevated risk of permanent injury from the infection. “Normal risk” refers to individuals without those specific known factors and also refers to individuals with risk factors that were not identified or were excluded in our analysis. High risk factors, by infection, include: measles—insufficient vitamin A; rubella—woman who had not contracted rubella before pregnancy; polio—absence of tonsils and not resting after the onset of significant symptoms; *Haemophilus influenzae* type b (Hib)—breastfed for <13 weeks; hepatitis B— infant of an infected mother, dwelling with an infected individual, sex with an infected partner, sex with multiple partners, men having sex with men, injection-drug use, and dwelling in a community with an unusually large group of infected individuals; and human papillomavirus (HPV)—smoking, women not screened every 3 years, and men with ≥6 oral sex partners in their lifetime.

**Table 2 vaccines-08-00561-t002:** Age groups that comprised the greatest proportion of deaths and permanent disabilities from various infectious diseases in the USA in the absence of mass vaccination.

Infection	Age Group	Proportion of Deaths and Permanent Disabilities
**Measles**	<10	91%
**Mumps**	<30	59%
**Rubella**	in utero	88%
**Tetanus**	<20	53%
**Diphtheria**	1–9	78%
**Pertussis**	<1	71%
**Polio**	<15	54%
**Hib**	<5	99%
**Hepatitis B**	≥50	79%
**Varicella**	≥20	54%
**HPV**	≥50	86%

**Table 3 vaccines-08-00561-t003:** Risk factors for elevated exposure to hepatitis B.

Age Group	Risk Factors
Children	Being born to a chronically infected mother, living with a chronically infected individual, and dwelling in a community that has a large number of infected individuals
Adults and adolescents	Having multiple sex partners, men having sex with men, injection-drug use, and dwelling in a community that has a large number of infected individuals

**Table 4 vaccines-08-00561-t004:** Types of HPV-attributable cancer and factors that elevate the risk of dying from them.

Gender	HPV-Attributable Cancers	Risk Factors
Female	Cervix, vagina, vulva, anus, rectum, and oropharynx	No Pap or HPV screening every three years and smoking
Male	Oropharynx, penis, anus, and rectum	Smoking and having six or more oral sex partners in a lifetime

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
