# Peer review of "Measuring the Benefits of Mass Vaccination Programs in the United States"

_vaccines, 2020, doi:10.3390/vaccines8040561_

Round 1

Reviewer 1 Report

The review from Magno and Golomb is an important investigation of the benefits of mass vaccination programs in the United States. The authors attempt to extrapolate the morbidity and mortality from a collection of vaccine-preventable infections, if there was no mass vaccination. These are specifically for MMR, tetanus, diphtheria, pertussis, HIb, HepB, varicella and HPV. This is a laudable exercise and I congratulate the authors for their comprehensive choice. The paper is well-written and I do not have any issues with the predictive calculations; I am pleased to see that the authors accept the obvious limitations of such a study in predicting morbidity and mortality in the absence of vaccination. I do have some comments and suggestions that I hope will improve the manuscript:

  1. The authors need to make it very clear why they are selecting 2014 as the date for assessing their study data. The rationale for selecting 2014 is not made. Are there more recent data that can be accessed and added to the study, e.g. up to 2018?
  2. I understand the thinking about omitting Prevenar-preventable infection and meningococcal vaccine-preventable infection; but, I think these are far too important to omit and the authors ought to give us a separate paragraph, perhaps at the end of their focus on these vaccine-preventable infections. I am concerned that reading especially about pneumococcal infection, one might conclude that using the vaccine has made no difference to overall pneumonia cases (it may well not do, if you have hundreds of thousands of viral pneumonia cases annually), whereas in reality it has impacted on streptococcal pneumonia. We should not lose sight of this.
    As for meningococcal vaccines, the conjugate vaccines have been tremendously effective and this needs to be stated, including up-to-date for Bexsero Group B vaccine (for controlling outbreaks in the US; though I am not sure if you have it for routine immunization).
  3. Table 1: it would be good to see a row at the bottom of the paper to show the cumulative morbidity (20 million) and the cases (1209 for normal risk and 10709 for high risk). So that this is clear when we read the beginning of the discussion. Also, please make clear in the title that it is annual estimated rates (I presume).
  4. The authors should discuss that even with high morbidity, advances in critical care treatment have contributed to survival rates and essentially low CFRs.
  5. The authors should give some data on the efficacy of these vaccines – a Table perhaps, as well as CFR estimates (which are hidden in the supplementary).
  6. In sections 3.1, 3.2 onwards, the authors should state that During the reference years XXX, add ‘before the introduction of mass vaccination’, just to re-iterate.
  7. The authors should also comment in Discussion that prior to vaccine use, infections in general were falling dramatically and that these infectious diseases rank low in the WHO predictions in 2020 and for 2050 for causes of mortality (e.g. compared with NCDs, traffic accidents, etc) and low compared to other intractable global diseases.
  8. The authors should also comment on how remarkably efficient the human immune system is when coupled with improved health care, etc., in fighting these infectious diseases (as suggested by the comments that most of the population, for example, contract measles, mumps and other infections without issue, and that even with pertussis, the deaths are remarkably low).
  9. A note on the impact of antibiotics on reducing morbidity/mortality is worthwhile in Discusion.
  10. Tables 3 and 4 can be combined with subheading for the infection.
  11. Line 292-294: suggests that adult booster doses may be necessary?
  12. Supplementary data: nicely presented and informative. Could the authors provide Figures like eFig1/2/3 for polio, Hib, HepB and varicella?
  13. Final comment – I feel that the authors need to have a conclusions paragraph in which they robustly defend mass vaccination campaigns, particularly in the US, where your anti-vaxxers are particularly vocal. It needs to be stressed that despite the fall in infectious diseases over the past century, we still need vaccination to prevent 20 million cases and 12,000 deaths, as well as to reduce the burden on health services, hospitals, intensive care and the economic damage, and improve QUALYS. In fact QUALYS is not used in the review and should be mentioned.

Author Response

We appreciate the reviews and the constructive comments and suggestions they made. The following are the comments and edits we made in response.

Review 1

  1. The authors need to make it very clear why they are selecting 2014 as the date for assessing their study data. The rationale for selecting 2014 is not made. Are there more recent data that can be accessed and added to the study, e.g. up to 2018?

Response: In lines 38-54, we moved the third paragraph of the methods section to the beginning and elaborated on the reason for our choice of 2014 as the target date for our projections.

  1. I understand the thinking about omitting Prevenar-preventable infection and meningococcal vaccine-preventable infection; but, I think these are far too important to omit and the authors ought to give us a separate paragraph, perhaps at the end of their focus on these vaccine-preventable infections. I am concerned that reading especially about pneumococcal infection, one might conclude that using the vaccine has made no difference to overall pneumonia cases (it may well not do, if you have hundreds of thousands of viral pneumonia cases annually), whereas in reality it has impacted on streptococcal pneumonia. We should not lose sight of this.
    As for meningococcal vaccines, the conjugate vaccines have been tremendously effective and this needs to be stated, including up-to-date for Bexsero Group B vaccine (for controlling outbreaks in the US; though I am not sure if you have it for routine immunization).

Response: In lines 73-88, we elaborated more on the reasons for omitting analyses of meningcoccal and pneumococcal diseases.

  1. Table 1: it would be good to see a row at the bottom of the paper to show the cumulative morbidity (20 million) and the cases (1209 for normal risk and 10709 for high risk). So that this is clear when we read the beginning of the discussion. Also, please make clear in the title that it is annual estimated rates (I presume).

Response: In lines 96-99, we added a summary of the cumulative findings shown in Table 1 at the beginning of the results section.

  1. The authors should discuss that even with high morbidity, advances in critical care treatment have contributed to survival rates and essentially low CFRs.

Response: We tried to address the effect of advances in health care throughout the report, such as in the tetanus and pertussis sections, when we “account for significant improvements in health care.” Also, in the discussion section, in line 342, we emphasize the importance of accounting for “health care, and treatment of disease, reflected in improved case fatality rates in unvaccinated populations.”

  1. The authors should give some data on the efficacy of these vaccines – a Table perhaps, as well as CFR estimates (which are hidden in the supplementary).

Response: We tried to provide data showing the effect of vaccines on mortality in the methods section, lines 55-59: “Although <100% of the population is vaccinated and vaccines are <100% effective, mass vaccination programs have contributed to significantly reduced morbidity and mortality of infectious disease. There were less than two dozen deaths from diphtheria, tetanus, pertussis, polio, measles, mumps, rubella, Haemophilus influenzae type b (Hib), or varicella in 2014. Therefore, we counted all estimated cases projected to 2014 in the absence of mass vaccination as preventable.”

We also added a case fatality/disability rate column to Table 1.

  1. In sections 3.1, 3.2 onwards, the authors should state that During the reference years XXX, add ‘before the introduction of mass vaccination’, just to re-iterate.

Response: In lines 116, 133, 141, 158, 172, 191, 207, 219, 235, and 255, we added “before the introduction of mass vaccination” to remind the reader of the meaning of the “reference years.”

  1. The authors should also comment in Discussion that prior to vaccine use, infections in general were falling dramatically and that these infectious diseases rank low in the WHO predictions in 2020 and for 2050 for causes of mortality (e.g. compared with NCDs, traffic accidents, etc) and low compared to other intractable global diseases.

Response: In lines 295-298, we added a sentence commenting on the general decline in mortality for all infectious disease.

  1. The authors should also comment on how remarkably efficient the human immune system is when coupled with improved health care, etc., in fighting these infectious diseases (as suggested by the comments that most of the population, for example, contract measles, mumps and other infections without issue, and that even with pertussis, the deaths are remarkably low).

Response: In lines 299-300, we added a sentence commenting on the evident efficiency of the immune system when couple with improved health care.

  1. A note on the impact of antibiotics on reducing morbidity/mortality is worthwhile in Discusion.

Response: In line 301, we added a sentence commenting on the positive effect of antibiotics on disease outcomes.

  1. Tables 3 and 4 can be combined with subheading for the infection.

Response: Although we appreciate the value of combining Tables 3 and 4, we felt that leaving those tables separate might be less confusing, as the tables are each part of the sections discussing their respective infections.

  1. Line 292-294: suggests that adult booster doses may be necessary?

Response: In line 315, we introduced the possibility that booster doses of HPV vaccine might be necessary for it to prevent cancer.

  1. Supplementary data: nicely presented and informative. Could the authors provide Figures like eFig1/2/3 for polio, Hib, HepB and varicella?

Response: Thank you for the positive feedback. Those figures are not straighforward as pre-vaccine mortality data for those infections were not tracked robustly or were tracked for a relatively short time period before vaccine licensure.

  1. Final comment – I feel that the authors need to have a conclusions paragraph in which they robustly defend mass vaccination campaigns, particularly in the US, where your anti-vaxxers are particularly vocal. It needs to be stressed that despite the fall in infectious diseases over the past century, we still need vaccination to prevent 20 million cases and 12,000 deaths, as well as to reduce the burden on health services, hospitals, intensive care and the economic damage, and improve QUALYS. In fact QUALYS is not used in the review and should be mentioned.

Response: In lines 348-352, we added a conclusions section and elaborated more on the benefits of mass vaccination programs.

Reviewer 2 Report

The manuscript provides an excellent analysis of the benefits of the US mass vaccination program to several infectious diseases (both viral and bacterial). The authors relied on many sources for information, such as reports of the CDC and other federal organizations, and especially scientific journals. The latter makes this study very valuable, since a significant portion of data from government may be incomplete. The reviewer agrees with the author’s methodology of the analysis, and that some infections are excluded. The rational for such exclusion was well justified. Overall, the manuscript is well written and should be a good sole source for the reference on subject.

One mistype should be corrected, on lines 209 and 210, H. influenzae (not a capital letter)     

Author Response

One mistype should be corrected, on lines 209 and 210, H. influenzae (not a capital letter)
Response: In lines 222, 223, we corrected a typo by replacing “H. Influenzae” with “H. influenzae.”